# DISENTANGLING TRAINABILITY AND GENERALIZATION IN DEEP LEARNING

## ABSTRACT

A fundamental goal in deep learning is the characterization of trainability and generalization of neural networks as a function of their architecture and hyper-parameters. In this paper, we discuss these challenging issues in the context of wide neural networks at large depths where we will see that the situation simplifies considerably. To do this, we leverage recent advances that have separately shown: (1) that in the wide network limit, random networks before training are Gaussian Processes governed by a kernel known as the Neural Network Gaussian Process (NNGP) kernel, (2) that at large depths the spectrum of the NNGP kernel simplifies considerably and becomes "weakly data-dependent", and (3) that gradient descent training of wide neural networks is described by a kernel called the Neural Tangent Kernel (NTK) that is related to the NNGP. Here we show that in the large depth limit the spectrum of the NTK simplifies in much the same way as that of the NNGP kernel. By analyzing this spectrum, we arrive at a precise characterization of trainability and a necessary condition for generalization across a range of architectures including Fully Connected Networks (FCNs) and Convolutional Neural Networks (CNNs). In particular, we find that there are large regions of hyperparameter space where networks can only memorize the training set in the sense they reach perfect training accuracy but completely fail to generalize outside the training set, in contrast with several recent results. By comparing CNNs with- and without-global average pooling, we show that CNNs without average pooling have very nearly identical learning dynamics to FCNs while CNNs with pooling contain a correction that alters its generalization performance. We perform a thorough empirical investigation of these theoretical results and finding excellent agreement on real datasets.

## 1 INTRODUCTION

Machine learning models based on deep neural networks have attained state-of-the-art performance across a dizzying array of tasks including vision (Cubuk et al., 2019), speech recognition (Park et al., 2019), machine translation (Bahdanau et al., 2014), chemical property prediction Gilmer et al. (2017), diagnosing medical conditions Raghu et al. (2019), and playing games Silver et al. (2018). Historically, the rampant success of deep learning models has lacked a sturdy theoretical foundation; architectures, hyperparameters, and learning algorithms are more often than not selected by brute force search Bergstra & Bengio (2012) and heuristics Glorot & Bengio (2010). Recently, significant theoretical progress has been made on several fronts that have shown promise in making neural network design more systematic. In particular, in the infinite width (or channel) limit, the distribution of functions induced by neural networks with random weights and biases has been precisely characterized before, during, and after training.

The study of infinite networks dates back to seminal work by Neal (1994) who showed that the distribution of functions given by single hidden-layer networks with random weights and biases in the infinite-width limit are Gaussian Processes (GPs). Recently, there has been renewed interest in studying random, infinite, networks starting with concurrent work on "conjugate kernels" (Daniely et al., 2016; Daniely, 2017) and "mean-field theory" (Poole et al., 2016; Schoenholz et al., 2017). The former set of papers argued that the empirical covariance matrix of pre-activations became deterministic in the infinite-width limit and called this the conjugate kernel of the network while the latter papers studied the properties of these limiting kernels along with the kernel describing

distribution of gradients. In particular, it was shown that the spectrum of the conjugate kernel of wide fully-connected networks approached a well-defined, data-independent, limit when the depth exceeds a certain scale, $\xi$. Networks with $tanh$-nonlinearities (among other bounded activations) exhibit a phase transition between two limiting spectral distributions of the conjugate kernel as a function of their hyperparameters with $\xi$ diverging at the transition. It was additionally hypothesized that networks were un-trainable when the conjugate kernel was sufficiently close to its limit.

Since then this analysis has been pushed to a wide range for architectures such as convolutions (Xiao et al., 2018), recurrent networks (Chen et al., 2018; Gilboa et al., 2019), networks with residual connections (Yang & Schoenholz, 2017), networks with quantized activations (Blumenfeld et al., 2019), the spectrum of the fisher (Karakida et al., 2018), a range of activation functions Hayou et al. (2018), and batch normalization (Yang et al., 2019). In each case, it was observed that the spectra of the kernels correlated strongly with whether or not the architectures were trainable. While these papers studied the properties of the conjugate kernels, especially the spectrum in the large-depth limit, a branch of concurrent work made a stronger statement: that many networks converge to Gaussian Processes as their width becomes large (Lee et al., 2018; Matthews et al., 2018; Novak et al., 2019b; Yang, 2019). In this case, the Conjugate Kernel was referred to as the Neural Network Gaussian Process (NNGP) kernel.

Together this work offered a significant advance to our understanding of wide neural networks; however, this theoretical progress was limited to networks at initialization or after Bayesian posterior estimation and provided no link to gradient descent. Moreover, there was some preliminary evidence that suggested the situation might be more nuanced than the qualitative link between the NNGP spectrum and trainability might suggest. For example, Philipp et al. (2017) showed that deep fully-connected $tanh$-networks could be trained after the kernel reached its large-depth, data-independent, limit but that these networks did not generalize to unseen data.

In the last year, significant theoretical clarity has been reached regarding the relationship between the GP prior and the distribution following gradient descent. In particular, Jacot et al. (2018) along with followup work (Lee et al., 2019; Chizat et al., 2019) showed that the distribution of functions induced by gradient descent for infinite-width networks is a Gaussian Process with a particular compositional kernel known as the Neural Tangent Kernel (NTK). In addition to characterizing the distribution over functions following gradient descent in the wide network limit, the learning dynamics can be solved analytically throughout optimization.

In this paper, we leverage these developments and revisit the relationship between architecture, hyperparameters, trainability, and generalization in the large-depth limit for a variety of neural networks. In particular, we make the following contributions:

1. We compute the large-depth asymptotics of several quantities related to trainability, including the largest eigenvalue of the NTK, $\lambda_{\max}$, and the condition number $\kappa = \lambda_{\max}/\lambda_{\min}$, where $\lambda_{\min}$ is the smallest eigenvalue; see Table 1.

2. We introduce the *residual predictor* $\Delta^{(l)}$, namely the difference between the finite depth and infinite depth NTK predictions, which is related to the model's ability to generalize: the network fails to generalize if $\Delta^{(l)}$ is too small.

3. We show that the *ordered* and *chaotic* phases identified in Poole et al. (2016) lead to markedly different limiting spectra of the NTK, which further indicates that, as a function of depth, the optimal learning rates ought to decay exponentially, linearly and remain roughly a constant in the chaotic, order-to-chaos and ordered phases, respectively.

4. We examine the differences in the above quantities for fully-connected networks (FCNs) and convolutional networks (CNNs) with and without pooling and precisely characterize the effect of pooling to these quantities.

5. We provide substantial experimental evidence supporting these claims, includes experiments that densely vary the hyperparameters of FCNs and CNNs with and without pooling.

Together these results provide a complete, analytically tractable, and dataset-independent theory for learning in very deep and wide networks. In addition to being interesting in its own right our theory provides a strong test of the NTK theory. Finally, our results provides clarity regarding the observation that for linear networks the learning rate must be decreased linearly in the depth of the

| | FC/CNN-F, | CNN-P | |
| --- | --- | --- | --- |
| NTK | Ordered $\chi_1 < 1$ | Critical $\chi_1 = 1$ | Chaotic $\chi_1 > 1$ |
| $\lambda_{\max}^{(l)}$ | $mq^* - m\mathcal{O}(l\chi_1^l)$ | $\frac{md+2}{3d}lq^* + m\mathcal{O}(1)$ | $\mathcal{O}(\chi_1^l)/d$ |
| $\lambda_{\text{rest}}^{(l)}$ | $\mathcal{O}(l\chi_1^l)/d$ | $\frac{2}{3d}q^*l + \frac{1}{d}\mathcal{O}(1)$ | $\mathcal{O}(\chi_1^l)/d$ |
| $\kappa^{(l)}$ | $dmq^*\mathcal{O}(\chi_1^{-l}/l)$ | $\frac{md+2}{2} + dm\mathcal{O}(l^{-1})$ | $\to 1$ |
| $\Delta^{(l)}$ | $\mathcal{O}(l\chi_1^l)/d$ | $d\mathcal{O}(l^{-1})$ | $d\mathcal{O}(l(\chi_{c^*}/\chi_1)^l)$ |

Table 1: **Evolution of the NTK spectra and $\Delta^{(l)}$.** The NTKs of FCN and CNN-F are essentially the same and the scaling of $\lambda_{\max}^{(l)}$, $\lambda_{\text{rest}}^{(l)}$, $\kappa^{(l)}$, and $\Delta^{(l)}$ for these networks is written in black. Corrections to these quantities due to the addition of an average pooling layer with window size $d$ is written in blue.

network Saxe et al. (2013). Here, we note that this is true only for networks that are initialized *critically*, i.e. on the order-to-chaos phase boundary.

## 2 BACKGROUND

We summarize recent developments in the study of wide random networks. We will keep our discussion relatively informal; see (Lee et al., 2018; Matthews et al., 2018; Novak et al., 2019b) for a more rigorous version of these arguments. To simplify this discussion and as a warmup for the main text, we will consider the case of FCNs. Consider a fully-connected network of depth $L$ where each layer has a width $N^{(l)}$ and an activation function $\phi : \mathbb{R} \to \mathbb{R}$. In this work we will take $\phi = \text{erf}$ however, most of the results will hold for a wide range of non-linearities though specifics - such as the phase diagram - can vary substantially. For simplicity, we will take the width of the hidden layers to infinity sequentially: $N^{(1)} \to \infty, \dots, N^{(L-1)} \to \infty$. The network is parameterized by weights and biases that we take to be randomly initialized with $W_{ij}^{(l)}, b_i^{(l)} \sim \mathcal{N}(0,1)$ along with hyperparameters, $\sigma_w$ and $\sigma_b$ that set the scale of the weights and biases. Letting the pre-activations in layer $l$ due to an input $x$ be given by $z_i^{(l)}(x)$, the network is then described by the recursion,

$$z_i^{(l+1)}(x) = \frac{\sigma_w}{\sqrt{N^{(l)}}} \sum_j W_{ij}^{(l+1)} \phi(z_j^{(l)}(x)) + \sigma_b b_i^{(l+1)} \qquad 0 \le l \le L-1. \qquad (1)$$

Notice that as $N^{(l)} \to \infty$, the sum ends up being over a large number of random variables and we can invoke the central limit theorem to conclude that the $\{z_i^{(l+1)}\}_{i \in [N]}$ are i.i.d. Gaussian with zero mean. Given a dataset of $m$ points, the distribution over pre-activations can therefore be described completely by the covariance matrix between neurons in different inputs $\mathcal{K}^{(l)}(x, x') = \mathbb{E}[z_i^{(l)}(x)z_i^{(l)}(x')]$. Inspecting Equation 75, we see that $\mathcal{K}^{(l+1)}(x, x')$ can be computed in terms of $\mathcal{K}^{(l)}(x, x')$ as

$$\mathcal{K}^{(l+1)}(x, x') = \sigma_w^2 \mathbb{E}_{(z, z') \sim \mathcal{N}(0, \mathcal{K}^{(l)}(x, x'))}[\phi(z)\phi(z')] + \sigma_b^2 \equiv \sigma_w^2 \mathcal{T}(\mathcal{K}^{(l)}(x, x')) + \sigma_b^2. \qquad (2)$$

for $\mathcal{T}$, an appropriately defined operator from the space of positive semi-definite matrices to itself.

Equation 2 describes a dynamical system on positive semi-definite matrices $\mathcal{K}(x, x')$. It was shown in Poole et al. (2016) that fixed points, $\mathcal{K}^*(x, x')$, of these dynamics exist such that $\lim_{l \to \infty} \mathcal{K}^{(l)}(x, x') = \mathcal{K}^*(x, x')$ with $\mathcal{K}^*(x, x') = q^*[\delta_{x,x'} + c^*(1 - \delta_{x,x'})]$ independent of the inputs $x$ and $x'$. The values of $q^*$ and $c^*$ are determined by the hyperparameters, $\sigma_w$ and $\sigma_b$. However Equation 2 admits multiple fixed points (e.g. $c^* = 0, 1$) and the stability of these fixed points plays a significant role in determining the properties of the network. Generically, there are large regions of the $(\sigma_w, \sigma_b)$ plane in which the fixed-point structure is constant punctuated by curves, called phase transitions, where the structure changes.

The rate at which $\mathcal{K}(x, x')$ approaches or departs $\mathcal{K}^*(x, x')$ can be determined by expanding Equation 2 about its fixed point, $\delta\mathcal{K}(x, x') = \mathcal{K}(x, x') - \mathcal{K}^*(x, x')$ to find[1]

$$\delta\mathcal{K}^{(l+1)}(x, x') \approx \sigma_w^2 \dot{\mathcal{T}}(\mathcal{K}^*(x, x'))\delta\mathcal{K}^{(l)}(x, x') \tag{3}$$

which exhibits exponential convergence to - or divergence from - the fixed-point as $\delta\mathcal{K}^{(l)}(x, x') \sim \chi(x, x')^l$ where $\chi(x, x') = \sigma_w^2 \dot{\mathcal{T}}(\mathcal{K}^*(x, x'))$. Since $\mathcal{K}^*(x, x')$ does not depend on $x$ or $x'$ it follows that $\chi(x, x')$ will take on a single value, $\chi_{c^*}$, whenever $x \neq x'$. If $\chi_{c^*} > 1$ then the fixed point is unstable and, as discussed above, there will be another fixed point that becomes stable, if $\chi_{c^*} < 1$ then the fixed point is stable, and if $\chi_{c^*} = 1$ then the hyperparameters lie at a phase transition. As was shown in Poole et al. (2016), there is always a fixed-point at $c^* = 1$ whose stability is determined by $\chi_1$. This defines the order-to-chaos transition. Note, that $\chi_{c^*}$ can be used to define a depth-scale, $\xi_{c^*} = -1/\log(\chi_{c^*})$ that describes the number of layers over which $\mathcal{K}^{(l)}$ approaches $\mathcal{K}^*$.

This provides a precise characterization of the NNGP kernel at large depths. As discussed above, recent work (Jacot et al., 2018; Lee et al., 2019; Chizat et al., 2019) has connected the prior described by the NNGP with the result of gradient descent training using a quantity called the NTK. To construct the NTK, suppose we enumerate all the parameters in the fully-connected network described above by $\theta_\alpha$. The finite width NTK is defined by $\hat{\Theta}(x, x') = J(x)J(x')^T$ where $J_{i\alpha}(x) = \partial_{\theta_\alpha} z_i^L(x)$ is the Jacobian evaluated at a point $x$. The main result in Jacot et al. (2018) was to show that in the infinite-width limit, the NTK converges to a deterministic kernel $\Theta$ and remains constant over the course of training. As such, at a time $t$ during gradient descent training with an MSE loss, the expected outputs of an infinitely wide network, $\mu_t(x) = \mathbb{E}[z_i^L(x)]$, evolve as

$$\mu_t(X_{\text{train}}) = (\mathbf{Id} - e^{-\eta\Theta_{\text{train, train}}t})Y_{\text{train}} \tag{4}$$

$$\mu_t(X_{\text{test}}) = \Theta_{\text{test, train}}\Theta_{\text{train, train}}^{-1}(\mathbf{Id} - e^{-\eta\Theta_{\text{train, train}}t})Y_{\text{train}} \tag{5}$$

for train and test points respectively; see Section 2 in Lee et al. (2019). Here $\Theta_{\text{test, train}}$ denotes the NTK between the test inputs $X_{\text{test}}$ and training inputs $X_{\text{train}}$ and $\Theta_{\text{train, train}}$ is defined similarly. Since $\hat{\Theta}$ converges to $\Theta$, the gradient flow dynamics of real network also converge to the dynamics described by Equation 4 and Equation 5 (Jacot et al., 2018; Lee et al., 2019; Chizat et al., 2019; Yang, 2019; Arora et al., 2019; Huang & Yau, 2019). As the training time, $t$ tends to infinity we note that these equations reduce to $\mu(X_{\text{train}}) = Y_{\text{train}}$ and $\mu(X_{\text{test}}) = \Theta_{\text{test, train}}\Theta_{\text{train, train}}^{-1}Y_{\text{train}}$. Consequently we call the linear operator

$$P(\Theta) \equiv \Theta_{\text{test, train}}\Theta_{\text{train, train}}^{-1} \tag{6}$$

the "mean predictor" or "predictor" for short. In addition to showing that the NTK describes networks during gradient descent, Jacot et al. (2018) showed that the NTK could be computed in closed form in terms of $\mathcal{T}, \dot{\mathcal{T}}$, and the NNGP as,

$$\Theta^{(l+1)}(x, x') = \mathcal{K}^{(l+1)}(x, x') + \sigma_w^2 \dot{\mathcal{T}}(\mathcal{K}^{(l)})(x, x')\Theta^{(l)}(x, x'). \tag{7}$$

where $\Theta^{(l)}$ is the NTK for the pre-activations at layer-$l$.

## 3 METRICS FOR TRAINABILITY AND GENERALIZATION AT LARGE DEPTH

We begin by discussing the interplay between the conditioning of $\Theta_{\text{train,train}}$ and the trainability of wide networks. We can write Equation 4 in terms of the spectrum of $\Theta_{\text{train,train}}$ letting $\Theta_{\text{train,train}} = U^T D U$ as,

$$\tilde{\mu}_t(X_{\text{train}})_i = (\text{Id} - e^{-\eta\lambda_i t})\tilde{Y}_{\text{train},i} \tag{8}$$

where $\lambda_i$ are the eigenvalues of $\Theta_{\text{train,train}}$ and $\tilde{\mu}_t(X_{\text{train}}) = U\mu_t(X_{\text{train}})$, $\tilde{Y}_{\text{train}} = UY_{\text{train}}$ are the mean prediction and the labels respectively written in the eigenbasis of $\Theta_{\text{train,train}}$. If we order the eigenvalues such that $\lambda_0 \geq \cdots \geq \lambda_M$ then it has been hypothesized in e.g. Lee et al. (2019) that the maximum feasible learning rate scales as $\eta \sim 2/\lambda_0$ as we verify empirically in section 4. Plugging

---

[1]More precisely, one needs to consider the Jacobian of $\mathcal{T}$ as an operator from positive semi-definite matrices to positive semi-definite matrices. We refer the readers to Section B of Xiao et al. (2018) for more details.

this scaling for $\eta$ into Equation 8 we see that the smallest eigenvalue will converge exponentially at a rate given by $\kappa = \lambda_M/\lambda_0$ the conditioning number. It follows that if the conditioning number of the NTK associated with a neural network diverges then it will become untrainable and so we use $\kappa$ as a metric for trainability. We will see that at large depths, the spectrum of $\Theta_{\text{train,train}}$ typically features a single large eigenvalue, $\lambda_{\max}$, and then a gap that is large compared with the rest of the spectrum. We therefore will often refer to a typical eigenvalue in the bulk as $\lambda_{\text{rest}}$ and approximate the condition number as $\kappa = \lambda_{\max}/\lambda_{\text{rest}}$.

In the large-depth limit we will see that $\Theta^{(l)}$ converges to $\Theta^*$ independent of the data distribution. In this case $\Theta^*_{\text{test,train}}$ will be a rank-1 constant matrix. As such, the mean prediction defined by Equation 5 completely fails to generalize. We define the finite depth correction to the infinite depth predictor[2],

$$\Delta^{(l)} Y_{\text{Train}} \equiv \left( P(\Theta^{(l)}) - P(\Theta^*) \right) Y_{\text{Train}}. \tag{9}$$

By the triangle inequality, the generalization error is lower bounded by

$$\|P(\Theta^{(l)})Y_{\text{train}} - Y_{\text{test}}\|_2 \geq \|P(\Theta^*)Y_{\text{train}} - Y_{\text{test}}\|_2 - \|\Delta^{(l)}Y_{\text{train}}\|_2 . \tag{10}$$

$\|P(\Theta^*)Y_{\text{train}} - Y_{\text{test}}\|_2$ is a constant independent of the test inputs and Equation 10 is large if $\|\Delta^{(l)}Y_{\text{Train}}\|_2^2$ is too small. Therefore, a necessary condition for the network to generalize is that there exists some $\rho > 0$ such that

$$\|\Delta^{(l)}Y_{\text{Train}}\|_2 \geq \rho \|P(\Theta^*)Y_{\text{train}} - Y_{\text{test}}\|_2 . \tag{11}$$

As such, we use $\Delta^{(l)}$ as a metric for generalization in this paper.

Our goal is therefore to characterize the evolution of the two metrics $\kappa^{(l)}$ and $\Delta^{(l)}$ in $l$. We follow the methodology outlined in Schoenholz et al. (2017); Xiao et al. (2018) to explore the spectrum of the NTK as a function of depth. We will use this to make precise predictions relating trainability and generalization to the hyperparameters $(\sigma_w, \sigma_b, l)$. Our main results are summarized in Table 1 which describes the evolution of $\lambda_{\max}^{(l)}$ (the largest eigenvalue of $\Theta^{(l)}$), $\lambda_{\text{rest}}^{(l)}$ (the remaining eigenvalues), $\kappa^{(l)}$, and $\Delta^{(l)}$ in three different phases (ordered, chaotic, and the phase transition) and their dependence on $m$, the size of the training set, the choices of architectures: FCN, CNN-F (convolution with flattening) and CNN-P (convolution with pooling), and size, $d$, of the window in the pooling layer (which we always take to be the penultimate layer).

We give a brief derivation of these results in Section 4 followed by a more detailed discussion in the appendix. However, it is useful to first give a qualitative overview of the phenomenology. In the ordered phase, $\lambda_{\max}^{(l)} \to m(q^* - l\chi_1^l)$ and $\lambda_{\text{rest}}^{(l)} \to l\chi_1^l$. At large depths since $\chi_1 < 1$ it follows that $\kappa^{(l)} \to mq^*/(l\chi_1^l)$ and so the condition number diverges exponentially quickly. Thus, in the ordered phase we expect networks not to be trainable (or, specifically, the rate at which they learn will grow exponentially in their depth). The predictor scales as $l\chi_1^l$ which goes to zero at the same rate as the divergence of $\kappa^{(l)}$; thus, in the ordered phase networks fail to train and generalize simultaneously. By contrast in the chaotic phase we see that there is no gap between $\lambda_{\max}^{(l)}$ and $\lambda_{\text{rest}}^{(l)}$ and networks become perfectly conditioned and are trainable everywhere. However, in this regime we see that the predictor scales as $l(\chi_{c^*}/\chi_1)^l$. Since, by definition, in the chaotic phase $\chi_{c^*} < 1$ and $\chi_1 > 1$ it follows that $\Delta^{(l)} \to 0$ over a depth $\xi = -1/\log(\chi_{c^*}/\chi_1)$. In the chaotic phase networks fail to generalize at a finite depth but remain trainable indefinitely. Finally, notice that introducing pooling modestly augments the depth over which networks can generalize in the chaotic phase but reduces the depth in the ordered phase. We will explore all of these predictions in detail in section 5.

## 4    LARGE-DEPTH ASYMPTOTICS OF THE NNGP AND NTK

We now give a brief derivation of the results in table 1. To simplify the notation we will discuss fully-connected networks and then extend the results to CNNs with pooling (CNN-P) and without pooling (CNN-F). Details of these two cases can be found in the appendix. We will focus on the

---

[2]If $\Theta^{(l)}$ diverges to infinity, we define $P(\Theta^*) = \lim_{l\to\infty} P(\Theta^{(l)})$. If $\Theta^*_{\text{train, train}}$ is singular, we will add a diagonal regularizer $\sigma \mathbf{Id}$ into $\Theta^*_{\text{train, train}}$.

NTK here since Schoenholz et al. (2017); Xiao et al. (2018) contains a detailed description of the NNGP in this case. As in sec. 2, we will be concerned with the fixed points of $\Theta$ as well as the linearization of Equation 7 about its fixed point. Recall that the fixed point structure is invariant within a phase so it suffices to consider the ordered phase, the chaotic phase, and the critical line separately. In cases where a stable fixed point exists, we will describe how $\Theta$ converges to the fixed point. We will see that in the chaotic phase and on the critical line, $\Theta$ has no stable fixed point and in that case we will describe its divergence. As above, in each case the fixed points of $\Theta$ have a simple structure with $\Theta^* = p^*((1 - \hat{c}^*)\mathbf{Id} + \hat{c}^*\mathbf{1}\mathbf{1}^T)$. To simplify the forthcoming analysis, without a loss of generality, we assume the inputs are normalized to have variance $q^*$ [3]. As such, we can treat $\mathcal{T}$ and $\dot{\mathcal{T}}$, restricted on $\{\mathcal{K}^{(l)}\}_l$, as a point-wise functions, since

$$\mathcal{T}(\mathcal{K})(x, x') = \mathbb{E}\phi(u)\phi(v), \quad (u, v)^T \sim \mathcal{N}\left(0, \begin{bmatrix} q^* & \mathcal{K}(x, x') \\ \mathcal{K}(x, x') & q^* \end{bmatrix}\right). \tag{12}$$

Since the off-diagonal elements approach the same fixed point at the same rate, we use $q_{ab}^{(l)} \equiv \mathcal{K}^{(l)}(x, x')$ and $p_{ab}^{(l)} \equiv \Theta^{(l)}(x, x')$ to denote any off diagonal entry of $\mathcal{K}^{(l)}$ and $\Theta^{(l)}$ respectively. We will similarly use $q_{ab}^*$ and $p_{ab}^*$ to denote the limits, $\lim_{l\to\infty} q_{ab}^{(l)} = q_{ab}^* = c^* q^*$ and $\lim_{l\to\infty} p_{ab}^{(l)} = p_{ab}^* = \hat{c}^* p^*$. Using the above notation, Equation 7 and Equation 2 become

$$q_{ab}^{(l+1)} = \sigma_w^2 \mathcal{T}(q_{ab}^{(l)}) + \sigma_b^2 \qquad\qquad p_{ab}^{(l+1)} = q_{ab}^{(l+1)} + \sigma_w^2 \dot{\mathcal{T}}(q_{ab}^{(l)}) p_{ab}^{(l)} \tag{13}$$

$$q^{(l+1)} = q^* \qquad\qquad p^{(l+1)} = q^* + \sigma_w^2 \dot{\mathcal{T}}(q^*) p^{(l)}, \tag{14}$$

where $p^{(l)} \equiv \Theta^{(l)}(x, x)$ and $q^{(l)} = \mathcal{K}^{(l)}(x, x)$. In what follows, we split the discussion into three parts according to the values of $\chi_1 \equiv \sigma_\omega^2 \dot{\mathcal{T}}(q^*)$ recalling that in Poole et al. (2016); Schoenholz et al. (2017) it was shown that $\chi_1$ controls the fixed point structure.

### 4.1 THE CHAOTIC PHASE $\chi_1 > 1$:

The chaotic phase is so-named because $q_{ab}^*/q^* < 1$ so that similar inputs become more uncorrelated as they pass through the network. In this phase, the diagonal entries of $\Theta^{(l)}$ grow exponentially and the off-diagonal entries converge to a fixed value. Indeed, Equation 14 implies,

$$p^{(l+1)} = q^* + \chi_1 p^{(l)} \qquad \Longrightarrow \qquad p^{(l)} = q^* \frac{\chi_1^{l+1} - 1}{\chi_1 - 1}, \tag{15}$$

which diverges exponentially. To find the limit of the off-diagonal terms, define $\chi_c = \sigma_\omega^2 \dot{\mathcal{T}}(q_{ab}^*)$ which was shown to control convergence of the $q_{ab}^{(l)}$ and is always less than 1 (Schoenholz et al., 2017; Xiao et al., 2018). Let $l \to \infty$ in Equation 13, we find that

$$p_{ab}^* = \frac{q_{ab}^*}{1 - \sigma_\omega^2 \dot{\mathcal{T}}(q_{ab}^*)} = \frac{q_{ab}^*}{1 - \chi_c} < \infty. \tag{16}$$

The rate of convergence of $p_{ab}^*$ is $\mathcal{O}(l\chi_c^l)$ (see Section A in the appendix). Since the diagonal terms diverge and the off-diagonal terms are finite it follows that in very deep networks in the chaotic phase, $(p^{(l)})^{-1}\Theta^{(l)} \to \mathbf{Id}$. Thus, in the chaotic phase, the spectrum of the NTK for very deep networks approaches the diverging constant multiplying the identity. From Equation 4 this implies that optimization in the chaotic phase should be easy since $\kappa^{(l)} \to 1$ (provided numerical precision issues from the prefactor do not become problematic). However, computing the mean prediction on test points and noticing that $P(\Theta^*)Y_{\text{train}} = 0$ we find (see Section B for the derivation),

$$\Delta^{(l)} Y_{\text{train}} = P(\Theta^{(l)})Y_{\text{train}} \approx (p^{(l)})^{-1}\mathcal{O}(l\chi_c^l)Y_{\text{train}} \to \mathbf{0}. \tag{17}$$

It follows that in the chaotic phase the networks predictions on unseen data to converge to 0 exponentially quickly in the depth. Since Equation 17 decays like $\mathcal{O}(l(p^{(l)})^{-1}\chi_c^l)$, we expect the network fails to generalize after $\mathcal{O}(\xi_*)$ layers, where $\xi_* = -1/(\log \chi_c - \log \chi_1)$ [4].

---

[3] It has been observed in previous works Poole et al. (2016); Schoenholz et al. (2017) that the diagonals converge much faster than the off-diagonals for $\tanh$- or erf- networks.

[4] For simplicity, we ignore the polynomial correction in $l$.

In summary, for wide networks, in the chaotic phase as the depth increases optimization becomes increasingly easy but the generalization performance degrades and eventually the network fails completely away from the training set after $\mathcal{O}(\xi_*)$ layers. Therefore, in the chaotic phase, deep network memorizes the training data. We will confirm this prediction for both kernel prediction and neural network training in the experimental results; see Fig 3.

## 4.2  THE ORDERED PHASE $\chi_1 = \sigma_\omega^2 \dot{\mathcal{T}}(q^*) < 1$:

The ordered phase is defined by the stable fixed point with $q_{ab}^*/q^* = 1$; in this case, disparate inputs will end up converging to the same output at the end of the network. In the ordered phase, Equation 14 implies that all the diagonal entries of $\Theta$ converge to the same value,

$$p^{(l)} = q^* \frac{\chi_1^{l+1} - 1}{\chi_1 - 1} \quad \xrightarrow{l \to \infty} \quad p^* = q^* \frac{1}{1 - \chi_1} < \infty \tag{18}$$

However, as with the NNGP kernel, the off-diagonal terms of the NTK, $p_{ab}^{(l)}$, will also converge to the value on the diagonal, $p^*$. It follows that the limiting kernels have the form $\Theta^* = p^* \mathbf{1}\mathbf{1}^{\mathbf{T}}$ and $\mathcal{K}^* = q^* \mathbf{1}\mathbf{1}^{\mathbf{T}}$. Thus, the limiting kernels are highly singular and feature only one non-zero eigenvalue. Since the limit is singular, we must linearize the dynamics about the fixed point to gain insight into the limiting behavior of the network. To compute the corrections, let

$$\epsilon_{ab}^{(l)} = q_{ab}^{(l)} - q_{ab}^* \qquad\qquad \delta_{ab}^{(l)} = p_{ab}^{(l)} - p_{ab}^* \tag{19}$$
$$\epsilon^{(l)} = q^{(l)} - q^* \qquad\qquad \delta^{(l)} = p^{(l)} - p^* \tag{20}$$

The diagonal correction can be obtained directly from Equation 18 and we find that $\epsilon^{(l)} = 0$ and $\delta^{(l)} = \frac{\chi_1^{l+1}}{1-\chi_1} q^*$. To compute correction of the off-diagonals, we linearize the equation around the fixed point to find that asymptotically (see Section A),

$$\epsilon_{ab}^{(l)} \approx \chi_1^l \epsilon_{ab}^{(0)} \qquad\qquad \delta_{ab}^{(l)} \approx \chi_1^l \left[ \delta_{ab}^{(0)} + l \left( 1 + \frac{\chi_2}{\chi_1} p_{ab}^* \right) \epsilon_{ab}^{(0)} \right] \tag{21}$$

where $\chi_2 = \sigma_\omega^2 \ddot{\mathcal{T}}(q^*)$. While the NNGP and NTK feature the same exponential rate of convergence set by $\chi_1$, we see that terms in the off-diagonal terms of the NTK feature polynomial corrections. $\Theta^{(l)}$ has (approximately) two eigenspaces. The first eigenspace comes from the single non-zero eigenvalue at the fixed point and it is very close to the DC mode (i.e. all entries of the eigenvector are equal to 1) with eigenvalue

$$\lambda_{\max}^{(l)} \approx (m-1)(p^* - \delta_{ab}^{(l)}) + (p^* - \delta^{(l)}) \to mp^* = \frac{mq^*}{1 - \chi_1} \tag{22}$$

i.e. is the sum of one row, where $m$ is the size of the dataset. The second eigenspace comes from lifting the degenerate zero-modes when $l < \infty$ and it has dimension $(m-1)$ with eigenvalue $\lambda_{\text{rest}}^{(l)} \approx -\delta^{(l)} + \delta_{ab}^{(l)} = \mathcal{O}(l\chi_1^l) \to 0$, which goes to zero exponentially over depth $l$. The eigenvalues of $\mathcal{K}^{(l)}$ have a similar distribution with $\lambda_{\max}^{(l)} \approx mq^* - (m-1)\epsilon_{ab}^{(l)}$ and $\lambda_{\text{rest}}^{(l)} = O(\chi_1^l)$. Thus the conditioning number, $\kappa^{(l)}$, of both $\Theta^{(l)}$ and $\mathcal{K}^{(l)}$ diverges exponentially as $\mathcal{O}(\chi_1^{-l} l^{-1})$ and $\mathcal{O}(\chi_1^{-l})$ respectively. As discussed above, there is a polynomial correction in the conditioning number of the NTK that slightly improves its conditioning. Since $\Theta^*$ is singular, we insert a diagonal regularization term $\sigma \mathbf{Id}$ into $\Theta_{\text{train, train}}$ of the linear predictor Equation 6, where $\sigma$ is a positive constant independent from $l$ and $\chi_1$. We find $\Delta^{(l)} = \mathcal{O}_\sigma(l\chi_1^l)$; see Section B for the derivation. In summary, in the ordered phase, $\xi_1 = -1/\log \chi_1$ (for simplicity, we ignore the polynomial correction) governs both trainability and generalizability of the predictor.

## 4.3  THE CRITICAL LINE $\chi_1 = \sigma_\omega^2 \dot{\mathcal{T}}(q^*) = 1$

On the critical line both the diagonal and the off-diagonal terms of $\Theta^{(l)}$ diverge linearly in the depth while $\mathcal{K}^{(l)}$ converges to $q^* \mathbf{1}\mathbf{1}^T$. From Equation 14 we see immediately that the diagonal terms are given by $q^{(l)} = q^*$ and $p^{(l)} = lq^*$. To compute the correction of the off-diagonals, we keep the

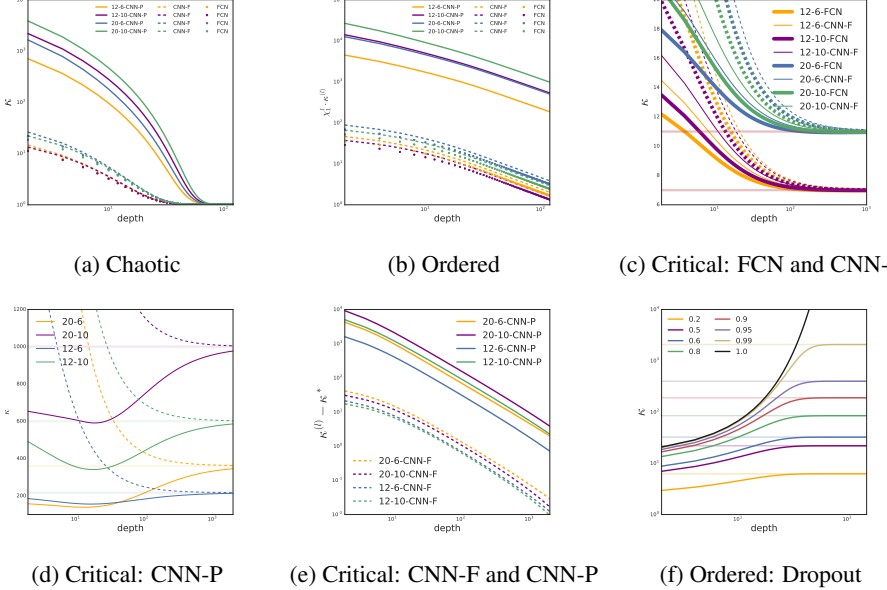

(a) Chaotic  (b) Ordered  (c) Critical: FCN and CNN-F

(d) Critical: CNN-P  (e) Critical: CNN-F and CNN-P  (f) Ordered: Dropout

Figure 1: **Condition numbers of NTKs and their rate of convergence.** Different colors represent images of different size. For example, in the yellow "12-6" , "12" represents the size of the dataset and "6" represents the dimension number ($6 * 6 * 3$ for FCN and $(6, 6, 3)$ for CNN) (a) In the chaotic phase, $\kappa^{(l)}$ converges to 1 for all architectures. (b) We plot $\chi_1^l \kappa^{(l)}$, confirming $\kappa$ explodes with rate $\chi_1^l/l$ in the ordered phase. In (c) and (d), the dashed lines representing the condition number $\kappa^{(l)}$ and solid lines the ratio between first and second eigenvalues. We see that, on the order-to-chaos transition, these two numbers converge to $\frac{m+2}{2}$ and $\frac{dm+2}{2}$ (horizontal lines) for FC/CNN-F and CNN-P respectively. In (e), we plot rates of convergence for CNN-P (solid) and CNN-F (dashed), confirming that pooling slows down the convergence of $\kappa^{(l)}$ by a factor of $d$. (f) Adding dropout to the penultimate layer prevents $\kappa^{(l)}$ from divergence in the ordered phase. The legends indicate the rate of the mask with $\rho = 1$ meaning keeping all activations. Horizontal lines are the limit of $\kappa^{(l)}$ computed in Equation 85 (here $m = 20$ for all curves.)
,

definition of $\epsilon_{ab}^{(l)}$ unchanged but define $\delta_{ab}^{(l)}$ slightly differently to the above as $\delta_{ab}^{(l)} = p_{ab}^{(l)} - lq^*$ to take into account the linear divergence at large depths. Taylor expanding to second order we find,

$$\epsilon_{ab}^{(l)} = -\frac{2}{\chi_2}\frac{1}{l} + o(\frac{1}{l}), \quad \delta_{ab}^{(l)} = -\frac{2}{3}lq^* + \mathcal{O}(1) \tag{23}$$

Thus for large $l$, $\Theta^{(l)}$ has the following form $p^{(l)} = lq^*$ and $p_{ab}^{(l)} = \frac{1}{3}lq^* + \mathcal{O}(1)$. As in the ordered phase, for large $l$ it follows that $\Theta^{(l)}$ essentially has two eigenspaces: one has dimension one and the other has dimension $(m - 1)$ with

$$\lambda_{\max}^{(l)} = \frac{(m + 2)q^*}{3}l + m\mathcal{O}(1), \quad \lambda_{\text{rest}}^{(l)} = \frac{2}{3}q^*l + \mathcal{O}(1) \tag{24}$$

and the condition number $\kappa^{(l)} = \frac{m+2}{2} + m\mathcal{O}(l^{-1}) \to \frac{m+2}{2}$ as $l \to \infty$. Unlike the chaotic and ordered phases, $\kappa^{(l)}$ converges with rate $\mathcal{O}(l^{-1})$. The $\mathcal{K}^{(l)}$ has $\lambda_{\max}^{(l)} = mq^* + m\mathcal{O}(l^{-1})$ and $\lambda_{\text{rest}}^{(l)} \approx \frac{2}{\chi_2}l^{-1}$ and the condition number $\kappa^{(l)}$ diverges linearly with slope $m\chi_2/2$. A similar calculation gives $\Delta^{(l)} = \mathcal{O}(l^{-1})$ on the critical line. In summary, $\kappa^{(l)}$ converges to a finite number and the network ought to be trainable for arbitrary depth but the residual predictor $\Delta^{(l)}$ decays polynomially, explaining why critically initialized networks with thousands of layers could still generalize (Xiao et al., 2018).

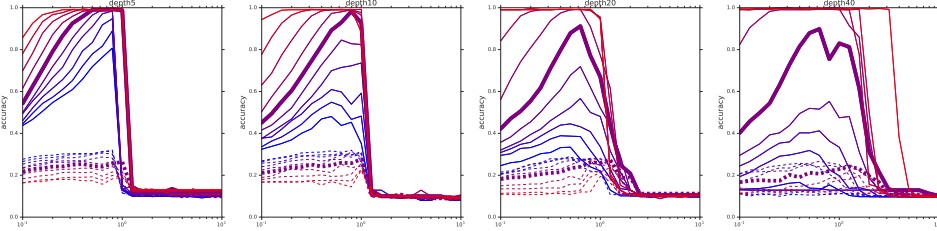

Figure 2: **Maximal learning rate can be calculated via the** $\lambda_{\max}$. $y$-axis: accuracy and $x$-axis: multiples of $\eta_{\text{theory}}$. Each point on the solid (dashed) lines represents the best training (test) accuracy throughout training of one configuration. From blue to purple to red, $(\sigma_\omega, \sigma_b)$ is moving from the order phase to the chaotic phase. $\rho = 1$ is the theoretical prediction.

### 4.4 Remarks

We end this section with a couple remarks. (1) The above theory holds for CNNs; see Section D. In the large depth setting, the NTK of CNNs without pooling is essentially the same as the NTK of FCNs. (2) In the ordered phase, Adding a dropout layer could significantly improve trainability of a network. For example, adding dropout to the penultimate layer, the condition number $\kappa^{(l)}$ will converge to a finite number rather than diverge exponentially; see (f) in Figure 1 and Equation 85 in the appendix.

## 5 Experiments

In this section, we provide empirical results to support the theoretical results in Section 4. Figure 1 is generated using synthetic data and all other plots are generated using CIFAR-10 with MSE as the loss function.

**Evolution of** $\kappa^{(l)}$ **(Figure 1).** We randomly sample inputs with shapes $(m, k^2 \times 3)$ for FCN and $(m, k, k, 3)$ for CNN-F/CNN-P, where $m \in \{12, 20\}$ and $k \in \{6, 10\}$. We compute the exact NTK with activation function *Erf* using the *Neural Tangents* library (Novak et al., 2019a). We see excellent agreement between the theoretical calculation of $\kappa^{(l)}$ in Section 4 (summarized in Table 1) and the experimental results Figure 1.

**Maximum Learning Rates (Figure 2 top).** In practice, given a set of hyper-parameters of a network, knowing the range of feasible learning rates is extremely valuable. As discussed above, in the infinite width setting, Equation 4 implies the maximal convergent learning rate is given by $\eta_{\text{theory}} \equiv 2/\lambda_{\max}^{(l)}$. We argue that $\eta_{\text{theory}}$ is a good prediction for the maximal convergent learning rate for wide network. To test this statement, we apply SGD to train a collection of fully-connected networks on CIFAR-10 using $1k$ training samples with the following configurations: (1) width: 2048 (2) $\sigma_b = 0.43$ fixed, (3) depths: $l = 5, 10, 20, 40$, (4) 10 different values of $\sigma_\omega$ moving from the ordered phase (blue) to the chaotic phase (red) (5) 10 different learning rates $\eta = \rho \eta_{\text{theory}}$, with $\rho \in [10^{-1}, 10^1]$. Overall, we see excellent agreement for depths less or equal to 20 and reasonable good agreement for depth 40. We point out that the degradation of the agreement for larger depth may due to the fact that the finite width NTK becomes more stochastic as the ratio between depth and width increases (Hanin & Nica, 2019). Note that Table 1 tells that, as depth increases, $\eta_{\text{theory}}$ should decays exponentially and linearly in the chaotic and critical phases resp. and remain roughly a constant in the ordered phase.

**Trainability vs Generalization (Figure 3 top).** Our theoretical result suggests that in the deep chaotic regime ($\chi_1^l$ is large) training becomes easier but the network can not generalize. On the other hand, the network can generalize but training becomes much more difficult as one moves towards the deep ordered region because $\kappa^{(l)}$ blows up exponentially. To confirm this claim, we conduct an experiment using 16k training samples from CIFAR-10 with $20 \times 20$ different $(\sigma_\omega, l)$ configurations. We train each network using SGD with batch size $b = 1024$ and learning rate $\eta = 0.3\eta_{\text{theory}}$. Deep in the chaotic phase we see that all configurations reach perfect training accuracy but the network

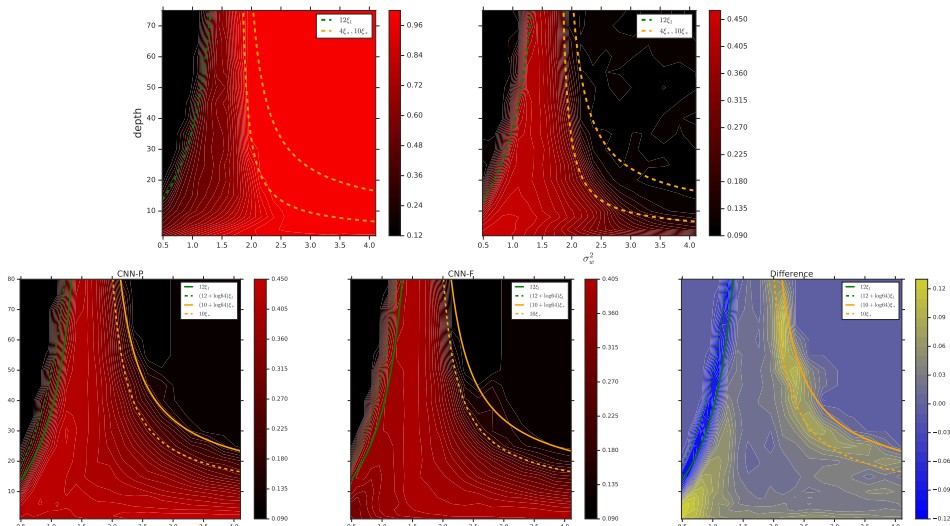

Figure 3: Top: training (left) and test accuracy of FCN using SGD. Bottom: test accuracy of CNN-P, CNN-F and the difference. In the blue strip, CNN-F significantly outperforms CNN-P, due to the fact that pooling increases the spectra gap by a factor of $d$.

completely fails to generalize in the sense test accuracy approaches $10\%$. However, in the ordered phase although the training accuracy degrades, generalization improves. The network eventually becomes untrainable after $\mathcal{O}(\xi_1)$ layers. In both phases we see that the depth scales, $\xi_1$ and $\xi_*$ respectively, perfectly capture the transition from generalizing to overfitting.

**CNN-P v.s. CNN-F: spatial correction (Figure 3 bottom).** We compute the test accuracy using the analytic NTK predictor Equation 5, which corresponds to the test accuracy of ensemble of gradient descent trained neural networks taking the width to infinity. We choose $1k$ training points, fix $\sigma_b^2$, and choose $20 \times 20$ different $(\sigma_\omega, l)$ configurations. We plot the test performance of CNN-P and CNN-F and the performance difference in Fig 3. Remarkably, the performance of both CNN-P and CNN-F are captured by $\xi_1 = -1/\log(\chi_1)$ in the ordered phase and by $\xi_* = -1/(\log \xi_c - \log \xi_1)$ in the chaotic phase. We see that the test performance difference between CNN-P and CNN-F exhibits a region in the ordered phase (a blue strip) where CNN-F outperforms CNN-P by a large margin. This performance difference is due to the correction term $d$ as predicted by the $\Delta^{(l)}$-row of Table 1.

## 6 CONCLUSION AND FUTURE WORK

In this work, we identify several quantities ($\lambda_{\max}$, $\lambda_{\text{rest}}$, $\kappa$, and $\Delta^{(l)}$) related to the spectrum of the NTK that control trainability and generalization of deep networks. We offer a precise characterization of these quantities and provide substantial experimental evidence supporting theoretical results. In future work, we would like to extend our framework to other architectures, e.g., ResNet (with batch-norm), attention model. Understanding the implication of the sub-Fourier modes in the NTK to the test performance of CNN is also an important research direction.

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

# A  SIGNAL PROPAGATION OF NNGP AND NTK

Recall that

$$q_{ab}^{(l+1)} = \sigma_w^2 \mathcal{T}(q_{ab}^{(l)}) + \sigma_b^2 \qquad\qquad p_{ab}^{(l+1)} = q_{ab}^{(l+1)} + \sigma_w^2 \dot{\mathcal{T}}(q_{ab}^{(l)}) \, p_{ab}^{(l)} \tag{25}$$

$$q^{(l+1)} = q^* \qquad\qquad p^{(l+1)} = q^* + \sigma_w^2 \dot{\mathcal{T}}(q^*) \, p^{(l)}, \tag{26}$$

## A.1  CORRECTION OF THE OFF-DIAGONALS IN THE CHAOTIC/ORDERED PHASE

Applying Taylor's expansion to the first equation of 25 gives

$$q_{ab}^* + \epsilon_{ab}^{(l+1)} = \sigma_\omega^2 \mathcal{T}(q_{ab}^* + \epsilon_{ab}^{(l)}) + \sigma_b^2 \tag{27}$$

$$= \sigma_\omega^2 \mathcal{T}(q_{ab}^*) + \sigma_b^2 + \sigma_\omega^2 \dot{T}(q_{ab}^*)\epsilon_{ab}^{(l)} + \mathcal{O}(\epsilon_{ab}^{(l)2}) \tag{28}$$

$$= q_{ab}^* + \sigma_\omega^2 \dot{T}(q_{ab}^*)\epsilon_{ab}^{(l)} + \mathcal{O}(\epsilon_{ab}^{(l)2}) \tag{29}$$

With $\chi_c = \sigma_\omega^2 \dot{\mathcal{T}}(q_{ab}^*)$, we have

$$\epsilon_{ab}^{(l+1)} \approx \chi_c \epsilon_{ab}^{(l)} \tag{30}$$

Similarly, applying Taylor's expansion to the second equation of 25 gives

$$\delta_{ab}^{(l+1)} \approx (1 + \chi_{c,2} p_{ab}^*)\epsilon_{ab}^{(l+1)} + \chi_c \delta_{ab}^{(l)} \tag{31}$$

where $\chi_{c,2} = \sigma_\omega^2 \ddot{\mathcal{T}}(q_{ab}^*)$. This implies

$$\epsilon_{ab}^{(l)} \approx \chi_c^l \, \epsilon_{ab}^{(0)} \tag{32}$$

$$\delta_{ab}^l \approx \chi_c^l \left[ \delta_{ab}^{(0)} + l \left( 1 + \frac{\chi_{c,2}}{\chi_c} p_{ab}^* \right) \epsilon_{ab}^{(0)} \right]. \tag{33}$$

Note that $\delta_{ab}^{(l)}$ contains a polynomial correction term and decays like $l\chi_c^l$. The correction to the fixed points in the ordered phase could be obtained using the same calculation:

$$\epsilon_{ab}^{(l)} \approx \chi_1^l \, \epsilon_{ab}^{(0)} \tag{34}$$

$$\delta_{ab}^{(l)} \approx \chi_1^l \left[ \delta_{ab}^{(0)} + l \left( 1 + \frac{\chi_2}{\chi_1} p^* \right) \epsilon_{ab}^{(0)} \right]. \tag{35}$$

## A.2  CORRECTION OF THE OFF-DIAGONALS ON THE CRITICAL LINE.

We have $\chi_1 = 1$ on the critical line. We need to expand the first equation of 25 to the second order

$$\epsilon_{ab}^{(l+1)} = \epsilon_{ab}^{(l)} + \frac{1}{2}\chi_2 \epsilon_{ab}^{(l)2} + \mathcal{O}(\epsilon_{ab}^{(l)3}) \tag{36}$$

Here we assume $\mathcal{T}$ has a continuous third derivative. The above equation implies

$$\epsilon_{ab}^{(l)} = -\frac{2}{\chi_2}\frac{1}{l} + o(\frac{1}{l}). \tag{37}$$

Then

$$\delta_{ab}^{(l+1)} = q_{ab}^{(l+1)} - q^* + \sigma_\omega^2 \dot{\mathcal{T}}(q^* + \epsilon_{ab}^{(l)})p_{ab}^{(l)} - lq^* \tag{38}$$

$$\approx \epsilon_{ab}^{(l+1)} + (\chi_1 + \chi_2 \epsilon_{ab}^{(l)} + \frac{1}{2}\chi_3(\epsilon_{ab}^{(l)})^2)(lq^* + \delta_{ab}^{(l)}) - lq* \tag{39}$$

$$\approx \epsilon_{ab}^{(l+1)} + (1 + \chi_2 \epsilon_{ab}^{(l)})\delta_{ab}^{(l)} + lq^* \chi_2 \epsilon_{ab}^{(l)} + \frac{1}{2}\chi_3(\epsilon_{ab}^{(l)})^2 lq^* \tag{40}$$

Plugging Equation 37 into the above equation gives

$$\delta_{ab}^{(l)} = -\frac{2}{3}lq^* + \mathcal{O}(1) \tag{41}$$

### A.3 RELU

We only consider the critical initialization $\sigma_\omega^2 = 2$ and $\sigma_b^2 = 0$. Using the equations in Appendix C of (Lee et al., 2019) gives

$$\sigma_\omega^2 \mathcal{T}(1 - \epsilon) = 1 - \epsilon + \frac{2\sqrt{2}}{3\pi}\epsilon^{3/2} + \mathcal{O}(\epsilon^{5/2}) \tag{42}$$

and taking the derivative w.r.t. $\epsilon$

$$\sigma_\omega^2 \dot{\mathcal{T}}(1 - \epsilon) = 1 - \frac{\sqrt{2}}{\pi}\epsilon^{1/2} + \mathcal{O}(\epsilon^{3/2}) \tag{43}$$

This is enough to conclude (similar to the above calculation)

$$\epsilon_{ab}^{(l)} = (\frac{3\pi}{\sqrt{2}})^2 l^{-2} + o(l^{-2}) \tag{44}$$

$$\delta_{ab}^{(l)} = -\frac{3}{4}l + \mathcal{O}(1). \tag{45}$$

Recall that $q^{(l)} = 1$ and $p^{(l)} = l$ for Relu network with $\sigma_\omega^2 = 2$. and $\sigma_b = 0$. Therefore

$$\lambda_{\max}^{(l)} = \frac{m+3}{4}l + m\mathcal{O}(1), \quad \lambda_{\text{rest}}^{(l)} = \frac{3}{4}l + \mathcal{O}(1) \quad \kappa^{(l)} = \frac{m+3}{3} + m\mathcal{O}(1/l). \tag{46}$$

### A.4 RESIDUAL RELU

We consider the following "continuum" residual network

$$x^{(t+dt)} = x^{(t)} + (dt)^{1/2}(W\phi(x^{(t)}) + b) \tag{47}$$

where $t$ denotes the number of layer and $dt > 0$ is sufficiently small and $W$ and $b$ are the weights and biases. We also set $\sigma_\omega^2 = 2$ (i.e. $\mathbb{E}[WW^T] = 2Id$) and $\sigma_b^2 = 0$ (i.e. $b = 0$). The NNGP and NTK have the following form

$$\mathcal{K}^{(t+dt)} = \mathcal{K}^{(t)} + 2dt\mathcal{T}(\mathcal{K}^{(t)}) \tag{48}$$

$$\Theta^{(t+dt)} = \Theta^{(t)} + dt\mathcal{K}^{(t)} + 2dt\dot{\mathcal{T}}(\mathcal{K}^{(t)})\Theta^{(t)} \tag{49}$$

Taking the limit $dt \to 0$ gives

$$\dot{\mathcal{K}}^{(t)} = 2\mathcal{T}(\mathcal{K}^{(t)}) \tag{50}$$

$$\dot{\Theta}^{(t)} = \mathcal{K}^{(t)} + 2\dot{\mathcal{T}}(\mathcal{K}^{(t)})\Theta^{(t)} \tag{51}$$

Using the fact that $q^0 = 1$ (i.e. the inputs have unit variance), we can compute the diagonal terms $q^{(t)} = e^t$ and $p^{(t)} = te^t$. Let $q_{ab}^{(t)} = e^t c_{ab}^{(t)}$, applying the above fractional Taylor expansion to $\mathcal{T}$ and $\dot{\mathcal{T}}$, we have

$$\dot{c}_{ab}^{(t)} = -\frac{2\sqrt{2}}{3\pi}(1 - c_{ab}^{(t)})^{\frac{3}{2}} + O((1 - c_{ab}^{(t)})^{\frac{5}{2}}) \tag{52}$$

Solving this gives

$$q_{ab}^{(t)} = (1 - \frac{9\pi^2}{2}t^{-2} + o(t^{-2}))e^t \tag{53}$$

$$p_{ab}^{(t)} = (\frac{1}{4}t + \mathcal{O}(1))e^t. \tag{54}$$

Thus the condition number is $m/3 + 1$. This is the same as the non-residual Relu case discussed above.

## B ASYMPTOTIC OF $\Delta^{(l)}$

To keep the notation simple, we denote $X_d = X_{\text{train}}$, $Y_d = Y_{\text{train}}$, $\Theta_{td} = \Theta_{\text{test, train}}$, $\Theta_{dd} = \Theta_{\text{train, train}}$. Recall that

$$\Delta^{(l)}Y_d = \left(P(\Theta^{(l)}) - P(\Theta^*)\right)Y_d = \left(\Theta_{td}^{(l)}\left(\Theta_{dd}^{(l)}\right)^{-1} - \Theta_{td}^*\left(\Theta_{dd}^*\right)^{-1}\right)Y_d \tag{55}$$

We split our calculation into three parts.

### B.1 Chaotic phase

In this case the diagonal $p^{(l)}$ diverges exponentially and the off-diagonals $p_{ab}^{(l)}$ converges to a bounded constant $p_{ab}^*$. Thus $P(\Theta^*)Y_d = \mathbf{0}$. We expand $\Theta^{(l)}$ about its fixe point

$$\Delta^{(l)}Y_d = \Theta_{td}^{(l)}\left(\Theta_{dd}^{(l)}\right)^{-1}Y_d \tag{56}$$

$$= \left(\Theta_{td}^* + \mathcal{O}(\delta_{ab}^{(l)})\right)\left(p^{(l)}\mathrm{Id} + p_{ab}^*(11^T - \mathrm{Id}) + \mathcal{O}(\delta_{ab}^{(l)})\right)^{-1}Y_d \tag{57}$$

$$= (p^{(l)})^{-1}\left(\Theta_{td}^* + \mathcal{O}(\delta_{ab}^{(l)})\right)\left(\mathrm{Id} - \frac{p_{ab}^*}{p^{(l)}}(11^T - \mathrm{Id}) + \mathcal{O}(\delta_{ab}^{(l)}/p^{(l)})\right)Y_d \tag{58}$$

$$= (p^{(l)})^{-1}\left(\Theta_{td}^* + \mathcal{O}(\delta_{ab}^{(l)})\right)\left(\mathrm{Id} - \frac{p_{ab}^*}{p^{(l)}}(11^T - \mathrm{Id}) + \mathcal{O}(\delta_{ab}^{(l)}/p^{(l)})\right)Y_d \tag{59}$$

$$= (p^{(l)})^{-1}\left(\mathcal{O}(\delta_{ab}^{(l)}) + \mathcal{O}(\delta_{ab}^{(l)}/p^{(l)})\right)Y_d \tag{60}$$

In the last equation, we have used the fact $11^T Y_d = \mathbf{0}$ and $\Theta_{td}^* Y_d = \mathbf{0}$ since $Y_d$ is balanced (i.e. containing the same number of positive (+1) and negative (-1) labels.) Therefore

$$\Delta^{(l)}Y_d\mathcal{O}((p^{(l)})^{-1}\delta_{ab}^{(l)}) = \mathcal{O}(l(\chi_c/\chi_1)^l). \tag{61}$$

### B.2 Order-to-chaos phase

Note that in this phase, both the diagonals and the off-diagonals diverge linearly. In this case

$$\lim_{l\to\infty}\frac{1}{lq^*}\Theta_{td}^{(l)} = \frac{1}{3}1_t1_d^T \quad \lim_{l\to\infty}\frac{1}{lq^*}\Theta_{dd}^{(l)} = B \equiv \frac{2}{3}\mathrm{Id} + \frac{1}{3}1_d1_d^T \tag{62}$$

Here we use $1_d$ to denote the all '1' (column) vector with length equal to the number of training points in $X_d$ and $1_t$ is defined similarly. Note that the constant matrix $B$ is invertible. By Equation 41

$$P(\Theta^{(l)}) = \frac{1}{3}\left(\frac{3}{lq^*}\Theta_{td}^{(l)}\right)\left(\frac{1}{lq^*}\Theta_{dd}^{(l)}\right)^{-1} \tag{63}$$

$$= \frac{1}{3}\left(1_t1_d^T + \mathcal{O}(1/lq^*)\right)(B + \mathcal{O}(1/lq^*))^{-1} \tag{64}$$

$$= \frac{1}{3}\left(1_t1_d^T + \mathcal{O}(1/lq^*)\right)\left(B^{-1} + \mathcal{O}(1/lq^*)\right) \tag{65}$$

$$= \frac{1}{3}1_t1_d^T B^{-1} + \mathcal{O}(1/lq^*) \tag{66}$$

$$= \lim_{l\to\infty}P(\Theta^{(l)}) + \mathcal{O}(1/lq^*) \tag{67}$$

Thus

$$\Delta^{(l)}Y_d = \mathcal{O}(1/lq^*) \tag{68}$$

Ordered Phase In this case $\Theta^*$ is a rank one matrix. We add a diagonal regularization term and defined

$$P_\sigma(\Theta) = \Theta_{td}\left(\Theta_{dd} + \sigma\mathrm{Id}\right)^{-1} \tag{69}$$

where $\sigma > 0$ is a positive constant independent of the hyper-parameters $(\sigma_w, \sigma_b, l)$. Let $B_\sigma = \Theta^* + \sigma\mathbf{Id}$. Then

$$P_\sigma(\Theta^{(l)}) = \left(\Theta_{td}^* + \mathcal{O}(\delta_{ab}^{(l)})\right)\left(B_\sigma + \mathcal{O}(\delta_{ab}^{(l)})\right)^{-1} \tag{70}$$

$$= \Theta_{td}^* B_\sigma^{-1} + \mathcal{O}_\sigma(\delta_{ab}^{(l)}) \tag{71}$$

$$= P_\sigma(\Theta^*) + \mathcal{O}_\sigma(\delta_{ab}^{(l)}) \tag{72}$$

## C  DROPOUT

In this section, we investigate the effect of adding a dropout layer to the penultimate layer. Let $0 < \rho \leq 1$ and $\gamma_j^{(L)}(x)$ be iid random variables

$$\gamma_j^{(L)}(x) = \begin{cases} 1, & \text{with probability} \quad \rho \\ 0, & \text{with probability} \quad 1 - \rho. \end{cases} \tag{73}$$

For $0 \leq l \leq L - 1$,

$$z_i^{(l+1)}(x) = \frac{\sigma_w}{\sqrt{N^{(l)}}} \sum_j W_{ij}^{(l+1)} \phi(z_j^{(l)}(x)) + \sigma_b b_i^{(l+1)} \tag{74}$$

and for the output layer,

$$z_i^{(L+1)}(x) = \frac{\sigma_w}{\rho\sqrt{N^{(L)}}} \sum_{j=1}^{N^{(L)}} W_{ij}^{(L+1)} \phi(z_j^{(L)}(x))\gamma_j^{(L)}(x) + \sigma_b b_i^{(L+1)} \tag{75}$$

where $W_{ij}^{(l)}$ and $b_i^{(l)}$ are iid Gaussians $\mathcal{N}(0,1)$. Since no dropout is applied in the first $L$ layers, the NNGP kernel $\mathcal{K}^{(l)}$ and $\Theta^{(l)}$ can be computed using Equation 2 and Equation 7. Let $\mathcal{K}_\rho^{(L+1)}$ and $\Theta_\rho^{(L+1)}$ denote the NNGP and NTK of the $(L+1)$-th layer. Note that when $\rho = 1$, $\mathcal{K}_1^{(L+1)} = \mathcal{K}^{(L+1)}$ and $\Theta_1^{(L+1)} = \Theta^{(L+1)}$ . We will compute the correction induced by $\rho < 1$. The fact

$$\mathbb{E}[\gamma_j^{(L)}(x)\gamma_i^{(L)}(x')] = \begin{cases} \rho^2, & \text{if} \quad (j,x) \neq (i,x') \\ \rho, & \text{if} \quad (j,x) = (i,x') \end{cases} \tag{76}$$

implies that the NNGP kernel $\mathcal{K}_\rho^{(L+1)}$ (Schoenholz et al., 2017) is

$$\mathcal{K}_\rho^{(L+1)}(x,x') \equiv \mathbb{E}[z_i^{(L+1)}(x)z_i^{(L+1)}(x')] = \begin{cases} \sigma_w^2 \mathcal{T}(\mathcal{K}^{(L)}(x,x')) + \sigma_b^2, & \text{if} \quad x \neq x' \\ \frac{1}{\rho}\sigma_w^2 \mathcal{T}(\mathcal{K}^{(L)}(x,x)) + \sigma_b^2 & \text{if} \quad x = x' . \end{cases} \tag{77}$$

Now we compute the NTK $\Theta_\rho^{(L+1)}$, which is a sum of two terms

$$\Theta_\rho^{(L+1)}(x,x') = \mathbb{E}\left[\frac{\partial z_i^{(L+1)}(x)}{\partial \theta^{(L+1)}} \left(\frac{\partial z_i^{(L+1)}(x')}{\partial \theta^{(L+1)}}\right)^T\right] + \mathbb{E}\left[\frac{\partial z_i^{(L+1)}(x)}{\partial \theta^{(\leq L)}} \left(\frac{\partial z_i^{(L+1)}(x')}{\partial \theta^{(\leq L)}}\right)^T\right]. \tag{78}$$

Here $\theta^{(L+1)}$ denote the parameters in the $(L+1)$ layer, namely, $W_{ij}^{(L+1)}$ and $b_i^{(L+1)}$ and $\theta^{(\leq L)}$ the remaining parameters. Note that the first term is in Equation 78 is equal to $\mathcal{K}_\rho^{(L+1)}(x,x')$. Using the chain rule, the second term is equal to

$$\frac{\sigma_\omega^2}{\rho^2 N^{(L)}}\mathbb{E}\left[\sum_{j,k=1}^{N^{(L)}} W_{ij}^{(L+1)} W_{ik}^{(L+1)} \dot{\phi}(z_j^{(L)}(x))\gamma_j^{(L)}(x)\dot{\phi}(z_k^{(L)}(x'))\gamma_j^{(L)}(x')\frac{\partial z_j^{(L)}(x)}{\partial \theta^{(\leq L)}} \left(\frac{\partial z_k^{(L)}(x')}{\partial \theta^{(\leq L)}}\right)^T\right] \tag{79}$$

$$= \frac{\sigma_\omega^2}{\rho^2 N^{(L)}}\mathbb{E}\left[\sum_j^{N^{(L)}} \dot{\phi}(z_j^{(L)}(x))\gamma_j^{(L)}(x)\dot{\phi}(z_j^{(L)}(x'))\gamma_j^{(L)}(x')\frac{\partial z_j^{(L)}(x)}{\partial \theta^{(\leq L)}} \left(\frac{\partial z_j^{(L)}(x')}{\partial \theta^{(\leq L)}}\right)^T\right] \tag{80}$$

$$= \frac{\sigma_\omega^2}{\rho^2}\mathbb{E}\left[\gamma_j^{(L)}(x)\gamma_j^{(L)}(x')\right]\mathbb{E}[\dot{\phi}(z_j^{(L)}(x))\dot{\phi}(z_j^{(L)}(x'))]\mathbb{E}\left[\frac{\partial z_j^{(L)}(x)}{\partial \theta^{(\leq L)}} \left(\frac{\partial z_j^{(L)}(x')}{\partial \theta^{(\leq L)}}\right)^T\right] \tag{81}$$

$$= \begin{cases} \sigma_\omega^2 \dot{\mathcal{T}}(\mathcal{K}^{(L)}(x,x'))\Theta^{(L)}(x,x') & \text{if} \quad x \neq x' \\ \frac{1}{\rho}\sigma_\omega^2 \dot{\mathcal{T}}(\mathcal{K}^{(L)}(x,x))\Theta^{(L)}(x,x) & \text{if} \quad x = x' . \end{cases} \tag{82}$$

In sum, we see that dropout only modifies the diagonal

$$
\begin{cases}
\Theta_\rho^{(L+1)}(x, x') = \Theta^{(L+1)}(x, x') \\[2mm]
\Theta_\rho^{(L+1)}(x, x) = \frac{1}{\rho}\Theta^{(L+1)}(x, x) + (1 - 1/\rho)\sigma_b^2
\end{cases}
\tag{83}
$$

In the ordered phase, we see

$$
\lim_{L\to\infty} \Theta_\rho^{(L)}(x, x') = p^*, \qquad \lim_{L\to\infty} \Theta_\rho^{(L)}(x, x) = \frac{1}{\rho}p^* + (1 - \frac{1}{\rho})\sigma_b^2
\tag{84}
$$

and the condition number

$$
\lim_{L\to\infty} \kappa_\rho^{(L)} = \frac{(m-1)p^* + \frac{1}{\rho}p^* + (1 - \frac{1}{\rho})\sigma_b^2}{(\frac{1}{\rho} - 1)(p^* - \sigma_b^2)} = \frac{mp^*}{(\frac{1}{\rho} - 1)(p^* - \sigma_b^2)} + 1
\tag{85}
$$

# D  CONVOLUTIONS

**General setup.** For simplicity of presentation we consider 1D convolutional networks with circular padding as in Xiao et al. (2018). We will see that this reduces to the fully-connected case introduced above if the image size is set to one and as such we will see that many of the same concepts and equations carry over schematically from the fully-connected case. The theory of two-dimensional convolutions proceeds identically but with more indices.

**Random weights and biases.** The parameters of the network are the convolutional filters and biases, $\omega_{ij,\beta}^l$ and $\mu_i^l$, respectively, with outgoing (incoming) channel index $i$ ($j$) and filter relative spatial location $\beta \in [\pm k] \equiv \{-k, \ldots, 0, \ldots, k\}$.[5] As above, we will assume a Gaussian prior on both the filter weights and biases,

$$
W_{ij,\beta}^l = \frac{\sigma_\omega}{\sqrt{(2k+1)n^l}}\omega_{ij,\beta}^l \qquad b_i^l = \sigma_b\mu_i^l, \qquad \omega_{ij,\beta}^l, \quad \mu_i^l \sim \mathcal{N}(0, 1)
\tag{86}
$$

As above, $\sigma_\omega^2$ and $\sigma_b^2$ are hyperparameters that control the variance of the weights and biases respectively. $n^l$ is the number of channels (filters) in layer $l$, $2k + 1$ is the filter size.

**Inputs, pre-activations, and activations.** Let $\mathcal{X}$ denote a set of input images. The network has activations $y^l(x)$ and pre-activations $z^l(x)$ for each input image $x \in \mathcal{X} \subset \mathbb{R}^{n^0 d}$, with input channel count $n^0 \in \mathbb{N}$, number of pixels $d \in \mathbb{N}$, where

$$
y_{i,\alpha}^l(x) \equiv \begin{cases} x_{i,\alpha} & l = 0 \\ \phi\left(z_{i,\alpha}^{l-1}(x)\right) & l > 0 \end{cases}, \qquad z_{i,\alpha}^l(x) \equiv \sum_{j=1}^{n^l} \sum_{\beta=-k}^{k} W_{ij,\beta}^l y_{j,\alpha+\beta}^l(x) + b_i^l.
\tag{87}
$$

$\phi : \mathbb{R} \to \mathbb{R}$ is a point-wise activation function. Since we assume circular padding for all the convolutional layers, the spacial size $d$ remains constant throughout the networks until the readout layer.

For each $l > 0$, as $\min\{n^1 \ldots, n^{l-1}\} \to \infty$, for each $i \in \mathbb{N}$, the pre-activation converges in distribution to $d$-dimensional Gaussian with mean $\mathbf{0}$ and covariance matrix $\mathcal{K}^{(l)}$, which can be computed recursively (Novak et al., 2019b; Xiao et al., 2018)

$$
\mathcal{K}^{(l+1)} = \mathcal{A} \circ \mathcal{T}(\mathcal{K}^{(l)}) = (\mathcal{A} \circ \mathcal{T})^{l+1}(\mathcal{K}^0)
\tag{88}
$$

Here $\mathcal{K}^{(l)} \equiv [\mathcal{K}_{\alpha,\alpha'}^{(l)}(x, x')]_{\alpha,\alpha'\in[d], x,x'\in\mathcal{X}}$, $\mathcal{T}$ is a non-linear transformation related to its fully-connected counterpart, and $\mathcal{A}$ a convolution coupled with a shift term acting on $\mathcal{X}d \times \mathcal{X}d$ PSD matrices

$$
[\mathcal{T}(\mathcal{K})]_{\alpha,\alpha'}(x, x') \equiv \mathbb{E}_{u\sim\mathcal{N}(0,\mathcal{K})}\left[\phi(u_\alpha(x))\phi(u_{\alpha'}(x'))\right]
\tag{89}
$$

$$
[\mathcal{A}(\mathcal{K})]_{\alpha,\alpha'}(x, x') \equiv \sigma_b^2 + \sigma_\omega^2 \sum_\beta \frac{1}{2k+1}[\mathcal{K}]_{\alpha+\beta,\alpha'+\beta}(x, x').
\tag{90}
$$

---

[5]We will use Roman letters to index channels and Greek letters for spatial location. We use letters $i, j, i', j'$, etc to denote channel indices, $\alpha, \alpha'$, etc to denote spatial indices and $\beta, \beta'$, etc for filter indices.

### D.1 THE NEURAL TANGENT KERNEL

To understand how the neural tangent kernel evolves with depth, we define the NTK of the $l$-th hidden layer to be $\hat{\Theta}^{(l)}$

$$\hat{\Theta}^{(l)}_{\alpha,\alpha'}(x,x') = \nabla_{\theta^{\leq l}} z^l_{i,\alpha}(x) \nabla_{\theta^{\leq l}} z^l_{i,\alpha'}(x') \tag{91}$$

where $\theta^{\leq l}$ denotes all of the parameters in layers at-or-above the $l$'th layer. It does not matter which channel index $i$ is used because as the number of channels approach infinity, this kernel will also converge in distribution to a deterministic kernel $\Theta^{(l+1)}$ (Yang, 2019), which can also be computed recursively in a similar manner to the NTK for fully-connected networks as,

$$\Theta^{(l+1)} = \mathcal{K}^{(l+1)} + \mathcal{A} \circ (\dot{\mathcal{T}}(\mathcal{K}^{(l)}) \odot \Theta^{(l)}) - \sigma_b^2, \tag{92}$$

where $\dot{\mathcal{T}}$ is given by Equation 89 with $\phi$ replaced by its derivative $\phi'$. We will also normalize the variance of the inputs to $q^*$ and hence treat $\mathcal{T}$ and $\dot{\mathcal{T}}$ as pointwise functions. We will only present the treatment in the chaotic phase to showcase how to deal with the operator $\mathcal{A}$. The treatment of other phases are similar. Note that the diagonal entries of $\mathcal{K}^{(l)}$ and $\Theta^{(l)}$ are exactly the same as the fully-connected setting. We only need to consider the off-diagonal terms. Letting $l \to \infty$ in Equation 92 we see that all the off-diagonal terms also converge . Let $\bar{\mathcal{A}} = \sigma_w^{-2}(\mathcal{A} - \sigma_b^2)$, be the normalized convolution operator. Note that $\mathcal{A}$ does not mix terms from different diagonals and it suffices to handle each off-diagonal separately. Let $\epsilon^{(l)}_{ab}$ and $\delta^{(l)}_{ab}$ denote the correction of the $j$-th diagonal of $\mathcal{K}^{(l)}$ and $\Theta^{(l)}$ to the fixed points. Linearizing Equation 88 and Equation 92 gives

$$\epsilon^{(l+1)}_{ab} \approx \chi_c \bar{\mathcal{A}} \epsilon^{(l)}_{ab} \tag{93}$$

$$\delta^{(l+1)}_{ab} \approx \chi_c \bar{\mathcal{A}} (\epsilon^{(l+1)}_{ab} + \frac{\chi_{c,2}}{\chi_c} p^*_{ab} \epsilon^{(l)}_{ab} + \delta^{(l)}_{ab}). \tag{94}$$

Next let $\{\rho_\alpha\}_\alpha$ be the eigenvalues of $\bar{\mathcal{A}}$ and $\epsilon^{(l)}_{ab,\alpha}$ and $\delta^{(l+1)}_{ab,\alpha}$ be the projection of $\epsilon^{(l+1)}_{ab}$ and $\delta^{(l)}_{ab}$ onto the $\alpha$-th eigenvector of $\bar{\mathcal{A}}$. Then for each $\alpha$,

$$\epsilon^{(l+1)}_{ab,\alpha} \approx (\rho_\alpha \chi_c)^{(l+1)} \epsilon^{(0)}_{ab,\alpha} \tag{95}$$

$$\delta^{(l)}_{ab,\alpha} \approx \rho_\alpha \chi_c (\epsilon^{(l+1)}_{ab,\alpha} + \frac{\chi_{c,2}}{\chi_c} p^*_{ab} \epsilon^{(l)}_{ab,\alpha} + \delta^{(l)}_{ab,\alpha}) \tag{96}$$

which gives

$$\delta^{(l)}_{ab} \approx \rho_\alpha \chi_c^l \epsilon_{ab,\alpha}, \quad \delta^{(l)}_{ab,\alpha} \approx \rho_\alpha^l \chi_c^l \left[ \delta^{(0)}_{ab,\alpha} + l \left( 1 + \frac{\chi_{c,2}}{\chi_c} p^*_{ab} \right) \epsilon^{(0)}_{ab,\alpha} \right] \tag{97}$$

Therefore, the correction $\Theta^{(l)} - \Theta^*$ propagates independently through different Fourier modes. In each mode, up to a scaling factor $\rho_\alpha$, the correction is the same as the correction of its FC counterpart. Since the subdominant modes (with $|\rho_\alpha| < 1$) decay exponentially faster than the dominant mode (with $\rho_\alpha = 1$), for large depth, the NTK of CNN is essentially the same as that of its FC counterpart.

### D.2 THE EFFECT OF POOLING AND FLATTENING OF CNNS

With the bulk of the theory in hand, we now turn our attention to CNNs. We show in the appendix that the dominant mode in CNNs behaves exactly like the fully-connected case, however we will see that the readout can significantly affect the spectrum. The NNGP and NTK of the $l$-th hidden layer CNN are 4D tensors $\mathcal{K}^{(l)}_{\alpha,\alpha'}(x,x')$ and $\Theta^{(l)}_{\alpha,\alpha'}(x,x')$, where $\alpha, \alpha' \in [d] \equiv [0, 1, \ldots, d-1]$ denote the pixel locations. To perform tasks like image classification or regression, "flattening" and "pooling" (more precisely, global average pooling) are two popular readout strategies that transform the last convolution layer into the logit layer. The former strategy "flattens" an image of size $(d, N)$ into a vector in $\mathbb{R}^{dN}$ and stacks a fully-connected layer on top. The latter projects the $(d, N)$ image into a vector of dimension $N$ via averaging out the spatial dimension and then stacks a fully-connected layer on top. The actions of "flattening" and "pooling" on the image correspond to computing the

mean of the trace and the mean of the pixel-to-pixel covariance on the NNGP/NTK, respectively, i.e.,

$$\Theta_{\text{flatten}}^{(l)}(x,x') = \frac{1}{d}\sum_{\alpha\in[d]}\Theta_{\alpha,\alpha}^{(l)}(x,x'), \quad \Theta_{\text{pool}}^{(l)}(x,x') = \frac{1}{d^2}\sum_{\alpha,\alpha'\in[d]}\Theta_{\alpha,\alpha'}^{(l)}(x,x') \tag{98}$$

where $\Theta_{\text{flatten}}^{(l)}$ ($\Theta_{\text{pool}}^{(l)}$) denotes the NTK right after flattening (pooling) the last convolution. We will also use $\Theta_{\text{fc}}^{(l)}$ to denote the NTK of FC. $\mathcal{K}_{\text{flatten}}^{(l)}$ and $\mathcal{K}_{\text{pool}}^{(l)}$ are defined similarly.

As discussed above, in the large depth setting, all the diagonals $\Theta_{\alpha,\alpha}^{(l)}(x,x) = p^{(l)}$ (since the inputs are normalized to have variance $q^*$ for each pixel) and similar to $\Theta_{\text{fc}}^{(l)}$, all the off-diagonals $\Theta_{\alpha',\alpha}^{(l)}(x,x')$ are almost equal (in the sense they have the same order of correction to $p_{ab}^*$ if exists.) Without loss of generality, we assume all off-diagonals are the same and equal to $p_{ab}^{(l)}$ (the leading correction of $q_{ab}^{(l)}$ for CNN and FCN are of the same order.) Applying flattening and pooling, the NTKs become

$$\Theta_{\text{flatten}}^{(l)}(x,x') = \frac{1}{d}\sum_{\alpha}\Theta_{\alpha,\alpha}^{(l)}(x,x') = \mathbf{1}_{x=x'}p^{(l)} + \mathbf{1}_{x\neq x'}p_{ab}^{(l)}, \tag{99}$$

$$\Theta_{\text{pool}}^{(l)}(x,x') = \frac{1}{d^2}\sum_{\alpha,\alpha'}\Theta_{\alpha,\alpha'}^{(l)}(x,x') = \frac{1}{d}\mathbf{1}_{x=x'}(p^{(l)} - p_{ab}^{(l)}) + p_{ab}^{(l)} \tag{100}$$

respectively. As we can see, $\Theta_{\text{ft}}^{(l)}$ is essentially the same as its FCN counterpart $\Theta_{\text{fc}}^{(l)}$ up to subdominant Fourier modes which decay exponentially faster than the dominant Fourier modes. Therefore the spectrum properties of $\Theta_{\text{ft}}^{(l)}$ and $\Theta_{\text{fc}}^{(l)}$ are essentially the same for large $l$.

However, pooling alters the NTK/NNGP spectrum in an interesting way. On the critical line, asymptotically, $\lambda_{\max}^{(l)} \approx (md+2)q^*l/(3d)$ and $\lambda_{\text{rest}}^{(l)} \approx 2q^*l/(3d)$, and $\kappa^{(l)} = \frac{md+2}{2} + md\mathcal{O}(l^{-1})$. Here we use blue color to indicate the changes of such quantities against their $\Theta_{\text{flatten}}^{(l)}$ counterpart. Thus pooling decreases $\lambda_{\text{rest}}^{(l)}$ roughly by a factor of $d$ and increases the condition number by a factor of $d$ comparing to flattening. In the chaotic phase, pooling does not change the off-diagonals $q_{ab}^{(l)} = \mathcal{O}(1)$ but does slow down the growth of the diagonals by a factor of $d$, i.e. $p^{(l)} = \mathcal{O}(\chi_1^l/d)$. This suggests, in the chaotic phase, there exists a transient regime of depths, where CNN-F hardly perform while CNN-P performs well. In the ordered phase, the pooling does not affect $\lambda_{\max}^{(l)}$ much but does decrease $\lambda_{\text{rest}}$ by a factor of $d$ and the condition number grows approximately like $d\chi_1^{-l}$, $d$ times bigger than its flattening and fully-connected network counterparts. This suggests the existence of a transient regime of depths, in which CNN-F outperforms CNN-P. This might be surprising because it is commonly believed CNN-P usually outperforms CNN-F.

