# OpenReview forum: "Disentangling Trainability and Generalization in Deep Learning"
_ICLR.cc/2020/Conference — Reject_

### Official Review · AnonReviewer2 · 2019-10-22
**Official Blind Review #2**

**Rating:** 3

**Review:**

This paper studies the relation between trainability and generalization ability in deep neural networks. In the theoretical analysis, the authors used the Neural network Gaussian process (NNGP) kernel and Neural Tangent kernel (NTK). The paper clarified that the spectrum of the NTK and NNGP has an important role in investigating the generalization and trainability, i.e., the condition number of the NTK. Some numerical experiments showed an agreement of theory with the practical behavior of learning algorithms.

In this paper, some existing theoretical results on deep neural networks were combined to extract new insight. Thought the attempt of this paper is interesting, the readability of the paper is not necessarily high.

- In equation (2), the operator T is defined as the kernel K(x,x'). However, the definition seems different from that in equation (8).  The authors need to make clear the definition of T.
- What is the "DC" mode in the sentence above the equation (15)?
- Is the derivation of the left part in equation (9) straightforward? How was the second term, chi_1 q^* p^(ell), derived?  I'm not sure how the dot{T} was dealt with. The argument below equation (3) should be used?

**Experience Assessment:**

I have read many papers in this area.

**Review Assessment: Checking Correctness Of Derivations And Theory:**

I did not assess the derivations or theory.

**Review Assessment: Checking Correctness Of Experiments:**

I did not assess the experiments.

**Review Assessment: Thoroughness In Paper Reading:**

I made a quick assessment of this paper.

---

> ### Author Response · Authors · 2019-11-15
> **Addressing major comments**
>
> Thank you for your careful review of our work. We appreciate your time and agree that our exposition needs to be improved.
>
> We have worked to make the exposition more friendly to newcomers and clearer to everyone. To do this we have made a few significant changes to our exposition. First, we have restricted the discussion in the main text to fully-connected networks and left a discussion of other architectures to the supplementary information. This allows us to improve the clarity of our exposition. We have also summarized the takeaway messages of our paper at the beginning and create a new table for our key results (Table 1 in the new version).
>
> ------------------------------------------------------------------------------------------------------------------------------
>
> "- In equation (2), the operator T is defined as the kernel K(x,x'). However, the definition seems different from that in equation (8).  The authors need to make clear the definition of T.
> "
> ------------------------------------------------------------------------------------------------------------------------------
>
> Thanks for pointing this out; equation 8 is correct and we have made the change.
>
>
> - What is the "DC" mode in the sentence above the equation (15)?
>
> ------------------------------------------------------------------------------------------------------------------------------
>
> Thanks for raising this. It represents the eigenvector whose entries are all equal to 1.
>
>
> "- Is the derivation of the left part in equation (9) straightforward? How was the second term, chi_1 q^* p^(ell), derived?  I'm not sure how the dot{T} was dealt with. The argument below equation (3) should be used? "
>
> ------------------------------------------------------------------------------------------------------------------------------
>
> Thanks. The main source of confusion comes from typos in the definition of $\chi_1$ in the paragraph below equation (8). The correction definition should be $\chi_1 = \sigma_w^2 \dot T(q^*)$. We have added more details to the deviation of this equation; see equations (13) and (14) in the new version.

---

### Official Review · AnonReviewer1 · 2019-10-23
**Official Blind Review #1**

**Rating:** 3

**Review:**

This paper studies the evolution of Neural Tangent Kernel (NTK) at large-depth regimes. By analyzing the conditional number and eigenvalues, they identify three phases of hyper-parameters; 1) In the chaotic phase NTK converges to an identity matrix, which is easy to train but hardly to generalize. 2) In the ordered phase NTK converges to an all-one matrix, which is hard to train but generalizes well. 3) In the critical phase the conditional number converges to a constant. Furthermore, they also analyze the influence of pooling and flattening in CNNs and identify potential regimes where pooling hurts the generalization. They conduct empirical experiments to supporting their theoretical analyses.

However, I think this paper is worth of more revisions because many theoretical analyses are unjustified. And some potential typos makes the analyses even more difficult to understand.

1) It looks to me that Eq(2) and Eq(6) are contradictory, where T already contains sigma_w and simga_b in Eq(2) but re-multiplied in eq(6).
2) The paper analyzes the dynamics by assuming the variances of inputs are q*, which is debatable. The variance q^l also evolves with the depth increases. It is unclear whether the condition number will change if you takes the evolution of q^l into considered.
3) It is unclear how Eq(9) comes directly from Eq(6), and there aren't any rigorous proofs in the Appendix. Similarly for eq(14).
4) In the paragraph below Eq(11) the paper states that \Theta* becomes an all one-matrix. However, Eq(11) states the diagonal converges to q*/(1-xi_1), but the paragraph below Eq(9) states the off-diagonal converges to q*_{ab}/(1- xi_c). Because q*=q*_{ab} as you stated nearby, do you mean xi_1 = xi_c ?
5) In the first paragraph of Section 3.3, p^l = q* and p^l=l q*.
6) In the 2nd contribution, you mentioned "eigenvector correlation", while I cannot find anywhere else introducing this.
7) The plots of Figure 1(b) should behave like convex if the kappa really evolves like x_1^l / l. However it is concave.
8) In the first experiment, you state "To confirm that the maximal feasible learning rate are ... 2/(lambda_max)". However, learning rates are never discussed in this paper. It is confusing why this experiment is useful.

Generally speaking, I think the paper needs careful revisions to support its theoretical analyses.

**Experience Assessment:**

I have read many papers in this area.

**Review Assessment: Checking Correctness Of Derivations And Theory:**

I assessed the sensibility of the derivations and theory.

**Review Assessment: Checking Correctness Of Experiments:**

I carefully checked the experiments.

**Review Assessment: Thoroughness In Paper Reading:**

I read the paper thoroughly.

---

> ### Author Response · Authors · 2019-11-15
> **Addressing major comments**
>
> Thank you for your extremely thorough reading of our paper. We appreciate your time and agree that our exposition needs to be improved. We have taken steps towards this in a round of revisions and will continue to improve the clarity of our writing. We believe our paper will be stronger as a result.
>
>
> 1) It looks to me that Eq(2) and Eq(6) are contradictory, where T already contains sigma_w and simga_b in Eq(2) but re-multiplied in eq(6).
> ------------------------------------------------------------------------------------------------------------------------------
>
> Thanks for pointing out the typos. The correct definition of $T$ does not contain $\sigma_b$, $\sigma_w$. Please see Eq(2) and (12) in the new version.
>
>
> 2) The paper analyzes the dynamics by assuming the variances of inputs are q*, which is debatable. The variance q^l also evolves with the depth increases. It is unclear whether the condition number will change if you takes the evolution of q^l into considered.
>
> ------------------------------------------------------------------------------------------------------------------------------
>
> In practice, the diagonal term $q^l$ converges much faster then the off-diagonal; see Figure 1 in Ben Pool etc. https://arxiv.org/pdf/1606.05340.pdf. This observation has been widely used in followup work to analyze the dynamics with q^l -> q* with excellent agreement [1,2,3]. Moreover, one can of course choose to normalize data so that the norm is exactly q* in the first layer. Nonetheless, we agree that we could do a better job of making this point in the text and have added a few sentences to this effect.
>
>
> 3) It is unclear how Eq(9) comes directly from Eq(6), and there aren't any rigorous proofs in the Appendix. Similarly for eq(14).
> ------------------------------------------------------------------------------------------------------------------------------
>
> Thanks for pointing out the gap between (6) and (9). The main source of confusion comes from typos in the definition of $\chi_1$ in the paragraph below equation (8). The correction definition should be $\chi_1 = \sigma_w^2 \dot T(q^*)$.  We have add new equations (Eq (13), (14) in the new version) to bridge the gap.
>
>
> 4) In the paragraph below Eq(11) the paper states that \Theta* becomes an all one-matrix. However, Eq(11) states the diagonal converges to q*/(1-xi_1), but the paragraph below Eq(9) states the off-diagonal converges to q*_{ab}/(1- xi_c). Because q*=q*_{ab} as you stated nearby, do you mean xi_1 = xi_c ?
>
> ------------------------------------------------------------------------------------------------------------------------------
>
> We could indeed improve the clarity of our discussion here. Note that in the ordered phase (where \Theta* becomes an all-ones matrix) it is indeed the case that \chi_c* == \chi_1 because c* = 1 is the only stable fixed-point.
>
> ------------------------------------------------------------------------------------------------------------------------------
>
>
> 5) In the first paragraph of Section 3.3, p^l = q* and p^l=l q*.
>
>
> Thanks. The first equation should be $q^l = q^*$.
>
>
> 6) In the 2nd contribution, you mentioned "eigenvector correlation", while I cannot find anywhere else introducing this.
>
> ------------------------------------------------------------------------------------------------------------------------------
>
> Thanks for pointing out this. We should have been more precise about this. We were referring to $Delta^l$. We have worked to clarify the exposition around $\Delta^l$ in general.
>
>
> 7) The plots of Figure 1(b) should behave like convex if the kappa really evolves like x_1^l / l. However it is concave.
>
> ------------------------------------------------------------------------------------------------------------------------------
>
> Thanks for bringing this up. To capture the polynomial correction, the Y-axis is indeed set to be $\chi_1^l \kappa^l$ (rather than $\kappa$), which should be roughly $1/l$ for large depth. We will make the labels more visible.
>
>
> 8) In the first experiment, you state "To confirm that the maximal feasible learning rate are ... 2/(lambda_max)". However, learning rates are never discussed in this paper. It is confusing why this experiment is useful.
>
> ------------------------------------------------------------------------------------------------------------------------------
>
> Thanks for pointing this out. We agree we should have clarified our interest in the maximum feasible learning rate. We have two reasons for investigating this point: 1) While it has been hypothesized in a number of recent papers that the maximum feasible learning rate scales like $\frac 2 {\lambda_{max}}$, we are not aware of a systematic study of this point. 2) In order to conduct subsequent experiments it was necessary to scale the learning rate appropriately since $\lambda_{max}$ varies by several orders of magnitude over the range of hyperparameters / depths that we study.

---

### Official Review · AnonReviewer3 · 2019-10-27
**Official Blind Review #3**

**Rating:** 3

**Review:**

This paper studies the spectra of neural tangent kernels (NTKs) at large depth -- first let width go to infinity, and then let depth go to infinity. At infinite depth the kernel has the form a*identity+b*(all-one matrix), and the paper studies how the large-depth NTK converges to the limit in three cases: chaotic, ordered, and critical line. The paper draws connection between these behaviors with the trainability and generalization of corresponding neural networks. Furthermore, the difference between CNNs with and without global average pooling is studied.

NTK has been a popular subject of research in deep learning theory, and it's an interesting direction to study the NTK in large depth. However, the exposition is confusing and I'm missing some key points of this paper. Therefore I cannot recommend acceptance at this time. See below for detailed comments.

1. I don't really get how the spectrum of large-depth NTK is connected to generalization. At infinite depth, the NTK is just a trivial kernel Theta^*, as noted in the paper. It is claimed that a finite-depth correction Eqn. (7) "captures the generalization." How exactly does it capture the generalization? Generalization appears to be highly dependent on the data distribution. I don't understand how the paper arrives at its conclusions regarding generalization.

2. The paper (esp. Section 3) is written in a way very unfriendly to someone who is not familiar with previous work, with notation, derivations and conclusions buried in paragraphs. I wish there were some theorems clearly and formally summarizing the conclusions.

3. It's unclear whether the studied regime (large depth, probably even larger with) is relevant in practice. Although there are experimental results provided, the CNN experiments are for the infinite-width NTK. It's unclear how they look like for practical networks.

4. There are numerous typos and grammar errors in the paper, even in abstract and introduction.


------
update:
Thanks to the authors for the response, especially the clarification about what they mean by generalization. Since the concern about the exposition is still present, I can only update my rating to "weak reject." I hope the authors could further improve the exposition of this paper.

**Experience Assessment:**

I have published one or two papers in this area.

**Review Assessment: Checking Correctness Of Derivations And Theory:**

I assessed the sensibility of the derivations and theory.

**Review Assessment: Checking Correctness Of Experiments:**

I assessed the sensibility of the experiments.

**Review Assessment: Thoroughness In Paper Reading:**

I made a quick assessment of this paper.

---

> ### Author Response · Authors · 2019-11-15
> **Addressing minor comments**
>
>
> 3. It's unclear whether the studied regime (large depth, probably even larger with) is relevant in practice. Although there are experimental results provided, the CNN experiments are for the infinite-width NTK. It's unclear how they look like for practical networks.
>
> ------------------------------------------------------------------------------------------------------------------------------------------------------
>
>
> Larger models often give superb performance, e.g. WideResNet, BERT https://arxiv.org/abs/1810.04805 (increasing the number of layers and heads), Efficient-Nets (increasing the resolution of the input images, widths and depths simultaneously) https://arxiv.org/abs/1905.11946. How to scale up the size of the models correctly and find the right range of hyper-parameters (weight/bias variances, learning rates, etc.) so that the models are able to train and generalize are important research problems in deep learning. Our paper gives insights into these questions: e.g. increasing the depth hurts generalization and trainability in the ordered phase, while in the chaotic phase, improve trainability but test performance degrades, showing a trade-off between generalization and optimization.  Note that previous works were not able to train deep networks in the chaotic phase because the learning rates are not chosen correctly. Understanding the effects of architecture modules (pooling, normalization, dropout, etc.) to generalization and trainability is critical for  architecture design. For example, we show that there is a trainability-generalization trade-off for `average pooling`: it increases the condition number of the NTK by $d$ (the window size of the pooling layer), but slow down the decay of $\Delta^l$ by $d$. We show that (new version of the paper), dropout improves the trainability of the network (or more precisely, the condition of the NTK) in the ordered phases and induces an implicit regularization that is similar to L2-regularization.
>
>
>
> 4. There are numerous typos and grammar errors in the paper, even in abstract and introduction.
>
> ------------------------------------------------------------------------------------------------------------------------------------------------------
>
> Thanks for pointing this out! We have worked to improve our exposition and cleaned up the writing.

---

> ### Author Response · Authors · 2019-11-15
> **Addressing major comments**
>
> We thank you for your thorough review of our work. We agree with the overall comments that our exposition could be significantly clearer. We have taken initial steps in this direction and will continue to improve the clarity of our results.
>
> As you correctly note, in the general case the spectrum of the NTK is deeply connected to specific properties of the dataset. This makes it difficult to study generalization and trainability in the general case and so simplifications must be made. Some approaches such as [1] choose to make progress here by considering simple data manifolds such as the unit  hypercube or hypersphere. However, we note that another place where simplicity can emerge is in the large depth limit and we believe that we can use this limit to gain significant insight into the properties of networks described by the NTK.
>
> At large depth we show that Theta becomes simple in the sense that it can be written as a small correction to the limiting kernel. Moreover, the scaling of the correction with depth is independent of dataset and can be studied in general for classes of neural network architectures. We leverage this property to make general comparisons between fully-connected networks and convolutional networks with-and-without pooling.
>
>
> ------------------------------------------------------------------------------------------------------------------------------------------------------
>
> 1. I don't really get how the spectrum of large-depth NTK is connected to generalization. At infinite depth, the NTK is just a trivial kernel Theta^*, as noted in the paper. It is claimed that a finite-depth correction Eqn. (7) "captures the generalization." How exactly does it capture the generalization? Generalization appears to be highly dependent on the data distribution. I don't understand how the paper arrives at its conclusions regarding generalization.
>
> ------------------------------------------------------------------------------------------------------------------------------------------------------
>
> Thanks for bring this up.  We claim that this is a necessary condition for generalization. The linear operator $\Delta^l$, in the infinite width setting (where the NTK captures dynamics), measures the distance between the finite depth (denoted by $l$) and the fixed-point (i.e. $l=\infty$, data independent) prediction. We used this to lower bound the generalization error; see equation (9) in the new version. The norm of this operator decays exponentially in the ordered and chaotic phases, and polynomially on the critical line. A necessary condition for the generalization error to be small is $\Delta^l$ shouldn’t be too small.  Thus after $O(-\log \Delta^l)$ layers, the generalization error has to be large. Green and yellow lines in Figure 3 correspond to $-\log \Delta^l$ ~ constant. To pass these arguments from kernel to finite (large) width network, we applied results from Jacot, https://arxiv.org/abs/1806.07572; Lee, et al https://arxiv.org/pdf/1902.06720, etc, that the training dynamics of real network is $1/\sqrt n$ away from its linearization and converges to the infinite width (i.e. analytic NTK) dynamics as $n$, the width of the network, goes to infinity.
>
> We agree with the reviewer that any fine-grained analysis of generalization should take the data distribution, optimization method, etc. into account. However, finding a necessary condition for generalization of neural networks in terms of hyper-parameters are important research questions. We believe that the large depth setting allows us to study these questions in a uniquely systematic manner.
>
>
>
>
>
> 2. The paper (esp. Section 3) is written in a way very unfriendly to someone who is not familiar with previous work, with notation, derivations and conclusions buried in paragraphs. I wish there were some theorems clearly and formally summarizing the conclusions.
>
> ------------------------------------------------------------------------------------------------------------------------------------------------------
>
> We agree with this comment; thanks for the feedback! We have worked to make the exposition more friendly to newcomers and clearer to everyone. To do this we have made a few significant changes to our exposition. First, we have restricted the discussion in the main text to fully-connected networks and left a discussion of other architectures to the supplementary information. This allows us to improve the clarity of our exposition. We have also summarized the takeaway messages of our paper at the beginning and create a new table to summarize our key results (Table 1 in the new version).
>
> [1] Greg Yang, Hadi Salman,  A Fine-Grained Spectral Perspective on Neural Networks

---

### Decision · Program_Chairs · 2019-12-19

**Decision:**

Reject

**Comment:**

The paper investigates the trainability and generalization of deep networks as a function of hyperparameters/architecture, while focusing on wide nets of large depth; it aims to characterize regions of hyperparameter space where networks generalize well vs where they do not; empirical observations are demonstrated to support theoretical results. However, all reviewers agree that, while the topic of the paper is important and interesting, more work is required to improve the readability and clarify the exposition to support the proposed theoretical results.